# Punch Motion Curve in the Extrusion–Drawing Process to Obtain Circular Cups

Tsung-Chia Chen [1] , Shi-Xun Chen [1] and Cheng-Chi Wang [2,3,*]

1 Department of Mechanical Engineering, National Chin-Yi University of Technology, Taichung 41170, Taiwan; ctchen@ncut.edu.tw (T.-C.C.); zxc24251960@gmail.com (S.-X.C.)
2 Department of Intelligent Automation Engineering, National Chin-Yi University of Technology, Taichung 41170, Taiwan
3 Graduate Institute of Precision Manufacturing, National Chin-Yi University of Technology, Taichung 41170, Taiwan
* Correspondence: wcc@ncut.edu.tw; Tel.: +886-4-23924505 (ext. 5163)

**Abstract:** Servo press technology is gaining attention because its punch motion curve offers greater formability than that of a conventional stamping press. This study investigated the effect of punch motion curves on the circular cup extrusion–drawing process. Various punch motion curves were analyzed, and the optimal curve for application was determined. Both the extrusion–drawing process and spring back of U-shaped sheet metal were investigated. In the circular cup extrusion–drawing process, the punch motion curve of a conventional stamping press (Case A) and three punch motion curves of a servo press (Cases B–D, the strokes of which differed from that of Case A by 0.5, 1.5, and 2.5 mm, respectively) were compared, particularly regarding the effect of the coefficient of friction on the circular cup extrusion–drawing process. The simulation analysis was performed using the software program DEFORM. A set of simulated parameters were compared with experimental results. The formability, cup shape, cup height, cup thickness, extrusion force–displacement curve, stress distribution, and strain distribution were analyzed for the design of the die required. Additionally, experimental and simulation results were compared to determine the reliability and precision of the DEFORM simulations. The results indicated that the conventional punch motion curve resulted in a shorter cup, greater stress, greater strain, and the need for a greater extrusion force. By contrast, the servo punch motion curves resulted in taller cups, less stress, and less strain. The findings can serve as a reference for the development of servo presses.

**Keywords:** servo press; motion curve; circular cup; extrusion

## 1. Introduction

The first steam-powered automobile was built by Nicolas-Joseph Cugnot in the late 17th century, and the first four-wheeled automobile was developed by Gottlieb Daimler in 1887. Since then, automobiles have become an indispensable means of transportation. After nearly two centuries, the automobile industry has fully matured, and the trend in automobile development is moving toward lighter, faster, and more streamlined models. With the advancement of automobile technology, attention is increasingly being paid to the appearance of automobiles and, thus, also to how sheet metal can be manipulated into the shapes required for streamlined automobile design. In manufacturing terms, this translates into a manufacturer's ability to massively produce sheet metal parts that meet tolerance requirements, which places a threshold on the punching performance of the manufacturer's stamping presses shown in Table 1. Such requirements have given rise to servo presses.

**Table 1.** The strengths and weaknesses of traditional methods.

| | Strengths | Weaknesses |
|---|---|---|
| traditional methods | Fast processing speed | 1. single stroke stamping presses<br>2. only one punch motion curve and the motion profiles cannot be adjusted<br>3. only move at a fixed speed for sheet metal stamping<br>4. wall thickness of the finished product is extremely uniform |

Servo presses overcome the limitation of conventional stamping presses: that they can have only one punch motion curve. When using a servo motor, the motion profiles of stamps can be freely adjusted. Servo motors are characterized by their position and velocity control abilities, with the velocity control range being wide. Additionally, the rotation speed of servo motors can be controlled with high precision. These properties have led to servo motors being widely employed in the machining of metal. Conventional stamping presses can be categorized into mechanical and hydraulic presses. Mechanical presses, which are mostly used in sheet metal stamping, are fast but can only move at a fixed speed. By contrast, hydraulic presses are capable of limited speed control, but they result in low production efficiency. Hydraulic presses can be further divided into oil presses and water presses, with oil presses being the predominant type and water presses being used only in specialized or large-size machinery. In servo presses, the stroke of the punch is adjustable, solving the problem that arises with conventional single stroke stamping presses. Servo presses have the benefits of a long die service life, high die precision, favorable machining capacity, low energy consumption, programmable control systems, and a freely adjustable punch motion curve (Figure 1) [1]; thus, they enable plastic deformation of sheet metal, giving it additional functionality.

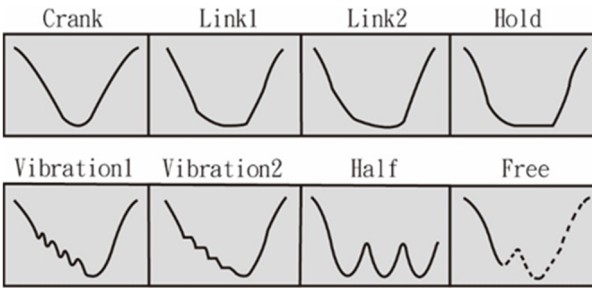

**Figure 1.** Various patterns of a punch motion curve [1].

Hu and Zu, in 1998, performed a finite element analysis to determine metal fluidity parameters for metal-shaping applications [2]. When extruding circular cups, blanks can rise in temperature over time, damaging the die; Hu and Zu thus modified the structure of a die to ensure it had sufficient strength. Srikanth and Murty (2012) used DEFORM to perform the finite element analysis on dies of various radii with a fixed friction coefficient and discovered that the ability to form circular cups was affected by the die radius [3]. They identified 7 mm as the optimal die radius for circular cup drawing. Atal and Shete, in 2013, employed FEM, a finite element analysis software program, to simulate the drawing of circular cups [4]. By simulating the predicted breaking time and the stress distribution of the circular cup's thickness, they could reduce the cost and time spent on the process substantially. Krishna and Swamy, in 2014, employed DEFORM to determine the optimal draw ratios for metal [5]. Their results indicated that in multistage drawing, the punch diameter in stage 1 (the draw) must be 40–50% of the blank diameter if the draw is to be successful; in stage 2, it must be 20–25% of the blank diameter in stage 1; and in stage 3, it must be 15–16% of the blank diameter in stage 2. Kanttikar, Kodli, and Chikmeti, in 2014,

simulated the deep-drawing process using a finite element analysis software program and reported that for dies with rounded angles, larger rounded angles resulted in smaller loads on the punch [6].

Olguner and Bozdana, in 2016, observed that the punch load, which was associated with the friction coefficient, increased with the length of a stroke and the flow of the blank in deep drawing [7]. After comparing various friction coefficients and making calculations using DEFORM, they concluded that the quality of circular cup formation decreased in the corners as the friction increased, resulting in thin areas or breakage. Olguner and Bozdana, in 2017, planned three sets of servo motion curves (frequency = 5, 10, and 20 Hz) for circular cup drawing and compared them with a conventional curve [8]. According to results obtained again using DEFORM, the punch force required for drawing decreased as the pulse frequency was increased, and when the motion curve was limited to the drawing of the circular cup, the formability of the circular cup was enhanced, and the chance of breakage was reduced. Swapna, Rao, and Radhika, in 2017, performed circular cup drawing with stainless steel, brass, and high-strength low-alloy steel [9]. These materials were selected because most studies have used aluminum or magnesium alloys. The researchers discovered that each material had its own characteristics and advantages in the drawing process. Bhatt and Buch, in 2017, developed a method for calculating the number of draws and die size required for circular cup drawing and were able to considerably shorten testing time and improve the production efficiency [10]. Chang and Yang, in 2017, employed DEFORM to simulate circular cup drawing and establish punch motion curves [11]. A comparison of their curves with those of conventional stamping presses revealed that the use of servo motion curves increased the blank diameter by 6–8%, enabling the processing to be improved and the desired product to be fabricated through only one draw. Furthermore, the degree of uniformity in blank size was found to be closely associated with the frequency of the servo motion curve.

In 2020, Kuo et. al. studied the SUS304 rectangular cup stamping and optimized the pulsating curve for a servo press using the finite element method. Meanwhile, they applied the Taguchi method to obtain the optimal parameter combinations and the optimization results showed that a shorter forming time (0.06 s less), a lower thinning ratio lower (0.1425% less), and a smaller forming force (808 N smaller) [12]. Kriechenbauer et al. proposed a systematic design of deep-drawing processes with free force and motion functions on servo presses based on computational science methods. They determined the optimal parameters (force and motion functions) for a deep-drawing process with superimposed vibrations on servo screw presses. The results of the evolutionary optimization approaches are validated by experiments with cross die part [13]. Choudhari et al. used the numerical and experimental approaches to analyze the effect of different drawing parameters such as blank shape, blank thickness, load, dry/wet lubrication on square cup drawing process for extra deep drawn steel sheet material. Simulation results are validated through experimentation. The optimized process parameters can be formed a square cup without any defects such as thinning, wrinkling, etc. The results showed that for considered process parameters, formability of material having a blank thickness of 2 mm is better as compared to a blank thickness of 1 mm and 0.8 mm, for load of 100 kN with dry lubrication [14].

In the present study, the servo curve method is proposed to show that as the frequency of the punch motion curve is the highest, a higher cup height can be obtained. The difference in the height of the cup during the extrusion–drawing process is only a slight change of 1.20%~2.12%. For the distribution of the thickness of the circular cup, the cup wall produced by the servo curve is the most uniform. If the cup wall obtained by the traditional curve is quite uniform, the servo curve method proposed in this paper will obtain the best extrusion–drawing effect. In this study, the circular cup extrusion–drawing process is designed according to the punch motion curve. In the future, the circular cup extension-drawing simulation analysis can be carried out, and a complete database of the circular cup extension–drawing process can be established to provide industrial production use to improve its production efficiency.

## 2. Material Properties and Experimental Design

### 2.1. SPCG-DC06 Sheet Steel

2.1.1. Material Parameters

The deep-drawing steel typically used in the automobile industry includes SPCD and SPCE sheet steel. This study used SPCG-DC06 sheets, a grade suitable for extra-deep drawing, to investigate the breakage of anisotropic steel sheets. These sheets are widely used in the automobile industry due to their excellent malleability, electrical conductivity, and thermal conductivity. In particular, SPCG sheets contain less C, Mn, P, and S than SPCD and SPCE sheets. Tables 2 and 3 present the chemical composition and material parameters, respectively, of the SPCG-DC06 sheets.

**Table 2.** Chemical composition of SPCG-DC06 sheets.

| C | Mn | P | S | Ti |
|---|---|---|---|---|
| 0.001 | 0.110 | 0.012 | 0.002 | 0.06 |

**Table 3.** Material parameters of SPCG-DC06 sheets.

| Material Parameters | Values |
|---|---|
| Tensile strength (MPa) | 270~330 |
| Tensile strength < (MPa) | 170 |
| Elongation > (%) | 41 |
| Plastic strain ratio > (r90) | 2.1 |
| Strain hardening exponent > n90 | 0.24 |

2.1.2. Tensile Test

This study used SPCG-DC06 sheets that were 2 mm in thickness in the process of circular cup extrusion. Figure 2 illustrates the setting for the tensile test and test pieces used, respectively. The material parameters had to be measured before the finite element method could be applied to simulate the circular cup extrusion–drawing process. A tensile test was conducted to obtain force, displacement, and time data, which were used for the calculation of engineering stress (s) and engineering strain (e) through Equations (1) and (2), respectively. Subsequently, Equations (3) and (4) were employed to convert engineering stress and engineering strain into the true stress ($\sigma$) and true strain ($\varepsilon$), respectively. Equation (5) was used to calculate the flow stress after relevant data were imported into DEFORM (Table 4). Figure 3 displays the anisotropic flow stress–strain rate and DC06 tensile test results, respectively.

$$s = \frac{F}{A_o}, \tag{1}$$

$$e = \frac{\Delta L}{L_o}, \tag{2}$$

$$\sigma = \frac{F}{A} = s(1+e), \tag{3}$$

$$\varepsilon = \ln\left(\frac{L}{L_o}\right) = \ln(1+e), \tag{4}$$

$$\overline{\sigma} = K(\overline{\varepsilon}_0 + \overline{\varepsilon})^n \dot{\overline{\varepsilon}}^m \exp\left(\frac{\beta}{T_{abs}}\right), \tag{5}$$

where $K$ and $n$ are material constants.

The true stress–true strain curve was determined using Equation (5), where $K$ and $n$ were obtained through the least root mean square method.

The degree of material anisotropy R typically represents the level of influence exerted by the direction of sheet metal rolling in a tensile test. In this study, tensile test pieces were rolled at $0°$, $45°$, and $90°$, which corresponded to an R value of $r_{0°}$, $r_{45°}$, and $r_{90°}$, respectively. The

results of the anisotropy test are presented in Table 5; the normal anisotropy $\bar{r}$ (shown as a mean value) and planar anisotropy $\Delta r$ were obtained using Equations (6) and (7), respectively.

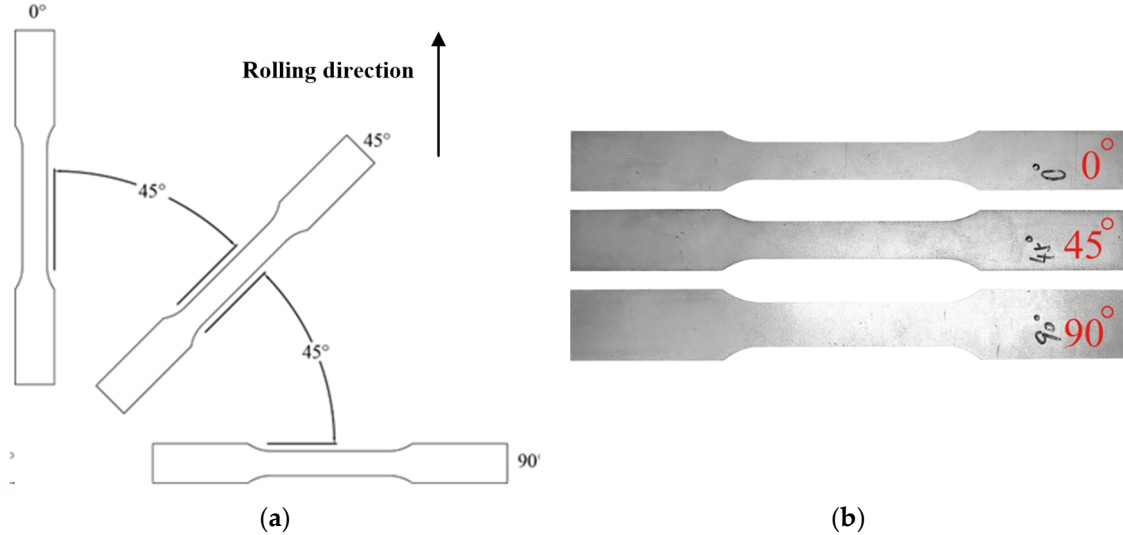

**Figure 2.** SPCG-DC06 sheets are used to be (**a**) the setting for the tensile test; (**b**) test pieces.

**Table 4.** Material parameters of SPCG-DC06 in DEFORM.

| Material thickness | E (GPa) | ν | K (MPa) | n | ε0 |
|---|---|---|---|---|---|
| 2 mm | 167 | 0.3 | 560.9 | 0.248 | 0.009 |

Where $E$: elastic factor; $v$: Poisson's ratio; $\sigma_y$: yield stress.

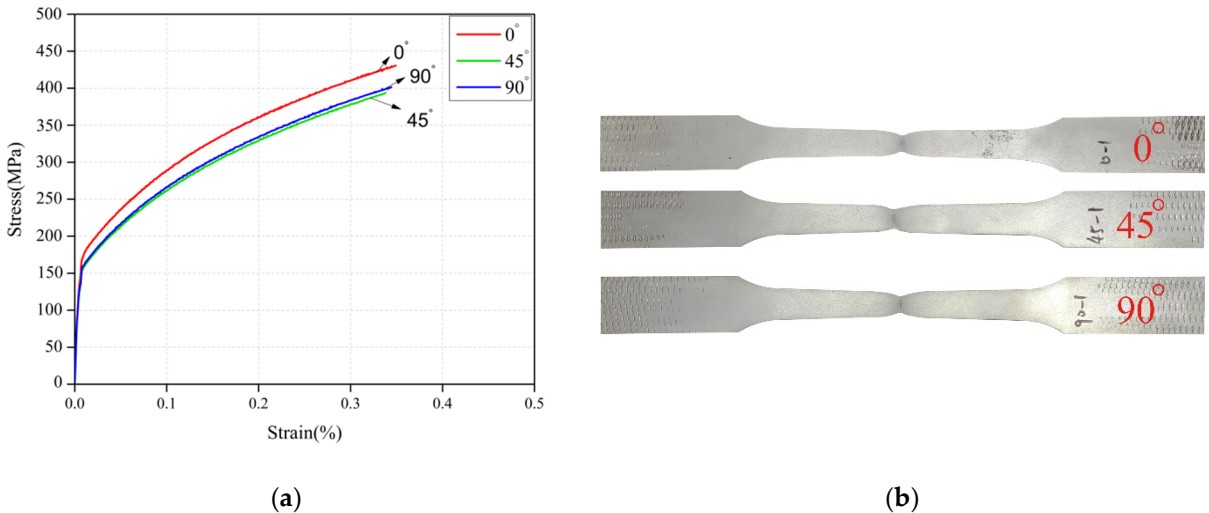

**Figure 3.** The experimental results for SPCG-DC06 sheets are obtained (**a**) stress–strain curve; (**b**) tensile test results of breakage.

**Table 5.** Results of the anisotropy test.

| Rolling Direction Direction | R | $\bar{r}$ | $\Delta r$ |
|---|---|---|---|
| $r_{0°}$ | 1.84 | | |
| $r_{45°}$ | 1.75 | 1.87 | 0.23 |
| $r_{90°}$ | 2.13 | | |

$$\bar{r} = \frac{r_{0°} + 2r_{45°} + r_{90°}}{4}, \tag{6}$$

$$\Delta r = \frac{r_{0°} - 2r_{45°} + r_{90°}}{2}, \tag{7}$$

### 2.2. Experimental Setting and Design of Die

The experiment was primarily conducted on an SD1-160 servo press shown in Figure 4, which performed the extrusion of circular cups. To determine the difference between experimental and simulation results, the extruded circular cups were measured. The cups were cut at angles $r_{0°}$, $r_{45°}$, and $r_{90°}$ by using an EXCETEK NP400L wire-cutting electrical discharge machine. The height and thickness of the cups cut at angles $r_{0°}$, $r_{45°}$, and $r_{90°}$ were measured using a Keyence VHX-5000 optical microscope at these angles. Because the bottom of the cups was rounded, which made conventional contact measurement difficult, a CWB-554A three-dimensional coordinate-measuring machine was employed. At the end of the extrusion process, the cups were rinsed in a DELTA DC150H ultrasonic cleaner, sectioned, embedded using a TOP TECH automatic mounting press, ground with a P20FS-1-R3 metallographic grinding and polishing machine, and cut into the desired size by using a high-precision diamond saw.

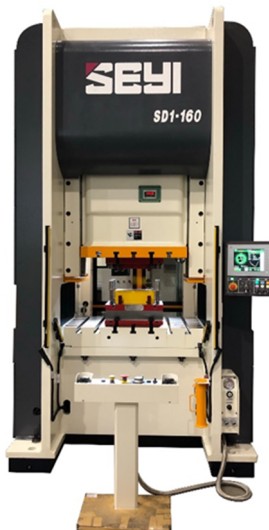

**Figure 4.** SD1-160 servo press.

The electronic servo press was paired with a set of dies for the extrusion of circular cups. A die, designed using Solidworks, was mounted in the servo press, as demonstrated in Figure 5a. The blanks were 51 mm in diameter and 2 mm in thickness; the punch was 24 mm in diameter with a corner radius of 3 mm; and the die was 26 mm in diameter with a corner radius of 5 mm. The die was made of SKD11. To shorten the processing time in the finite element analysis, a quarter-symmetric model was used. Figure 5b presents the dimensions of the quarter-symmetric model die.

### 2.3. Parameter and Device Setting for Circular Cup Extrusion Simulation

DEFORM was used to simulate the circular cup extrusion process. First, a motion curve (Case D) for the punch was developed (Figure 6), and the blank holder force was set at 900 N to ensure the smooth flow of the blank. The grid number was set at 140,000, and local grid refinement was implemented (Figure 7). The coefficient of friction was set to 0.1. Table 6 presents the parameters for the simulated extrusion–drawing process. The stresses induced when the punch moved down and up are displayed in Figure 8a,b, respectively.

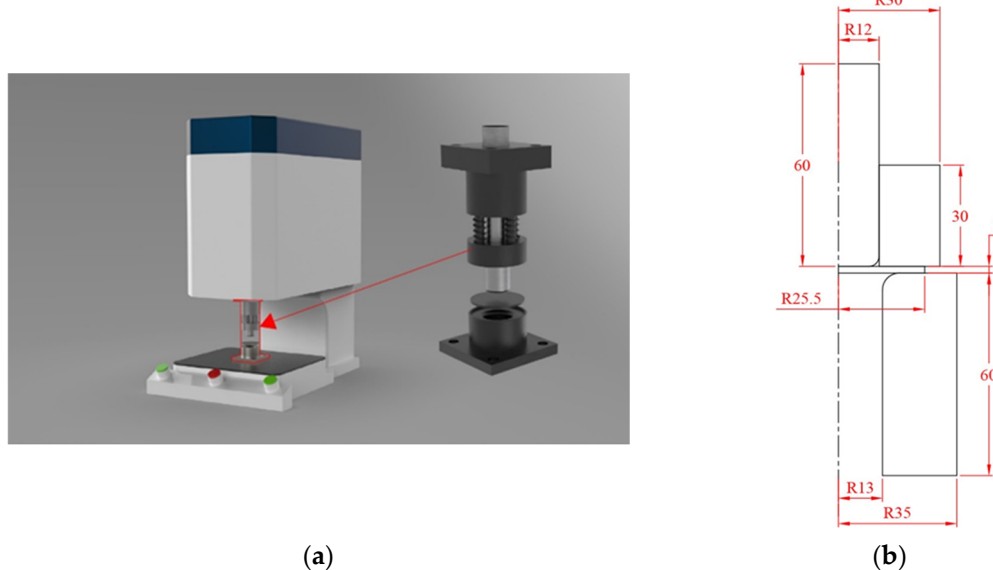

|  |  |
| :---: | :---: |
| (**a**) | (**b**) |

**Figure 5.** (**a**) Placement of the die in the servo press; (**b**) Dimensions of the quarter die.

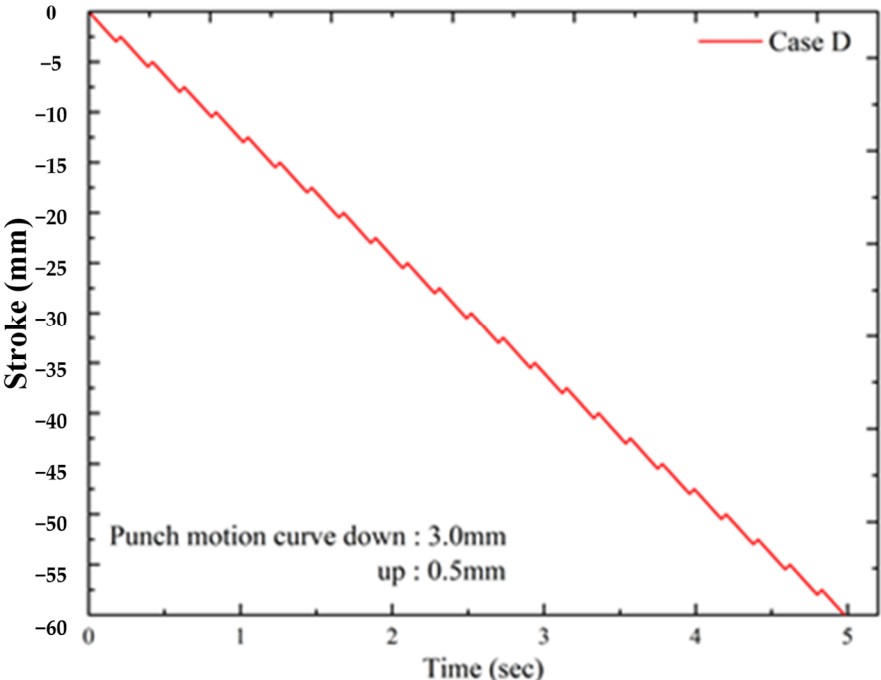

**Figure 6.** Punch motion curve (Case D).

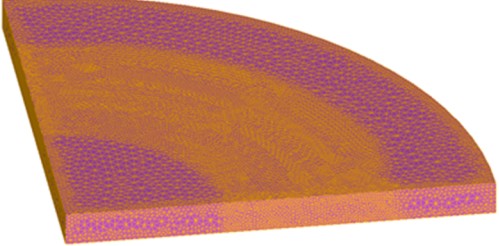

**Figure 7.** Local grid refinement.

**Table 6.** Parameters for the simulated extrusion process.

| Parameters | Values |
|---|---|
| Blank size (mm) | $\phi 51$ |
| Blank thickness (mm) | 2 |
| Punch corner radius (mm) | 3 |
| Die corner radius (mm) | 5 |
| Die clearance (mm) | 1 |
| Blank holder force (N) | 900 |
| Coefficient of friction | 0.1 |
| Number of grids | 140,000 |

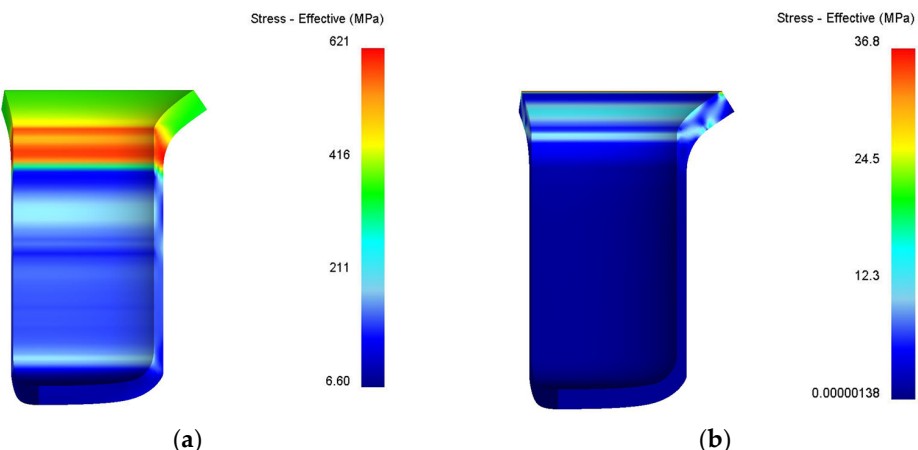

(a)  (b)

**Figure 8.** Stress when punch moved (**a**) down; (**b**) up.

## 3. Results and Discussions

### 3.1. Comparison of Experimental and Simulated Extrusion Results

Figure 9 presents a photograph of the circular cup produced using the electronic servo press with the punch motion curve for Case D. This cup was compared with the simulated cup; their differences are illustrated using the extrusion force–displacement, a cup shape diagram, and a thickness distribution diagram.

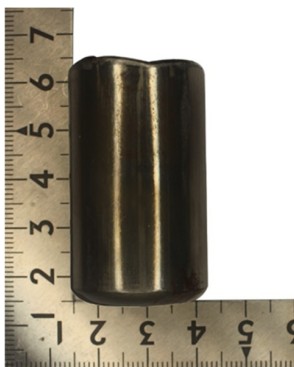

**Figure 9.** Circular cup produced using the punch motion curve for Case D.

#### 3.1.1. Extrusion Force–Displacement Diagram

The punch motion curve for Case D was applied to the extrusion of circular cups. Figure 10 presents the comparison of experimental and simulation results. The trends were largely the same before the stroke of the punch had reached 8 mm, and they remained similar between 8 and 33 mm with errors within acceptable engineering tolerance. In the final segment, the punch can be seen to have been subjected to a load. This could be attributed to the continual acquisition of signals when the punch was moving back.

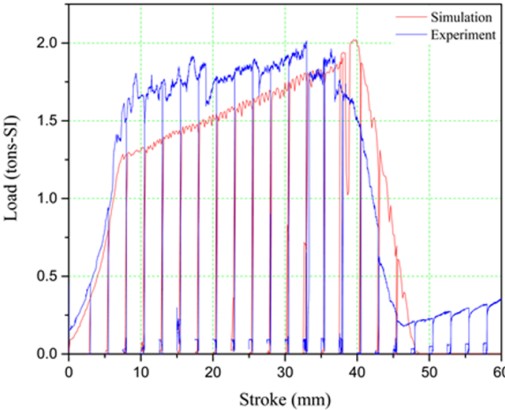

**Figure 10.** Extrusion force–displacement diagram when applying Case D.

3.1.2. Comparison of Extruded Cup Shape

Figure 11 presents a comparison of cup shapes in the experiment results ($r_{0°}$, $r_{45°}$, and $r_{90°}$) and simulation results (DEFORM). Figure 12 presents photographs of the cups cut at angles $r_{0°}$, $r_{45°}$, and $r_{90°}$ by using a wire-cutting electrical discharge machine. An optical microscope, laser displacement meter, and image-measuring instrument were employed in the shape comparison. For $r_{0°}$, the bottom of the cup was similar in shape to that of the simulated cup; the rounded corner deviated somewhat from the simulated corner, but the difference remained within acceptable engineering tolerance. Additionally, the experimental cup's wall was identical to the simulated cup's wall, with there being only an extremely small difference at the top of the cup. For $r_{45°}$, the bottom and rounded corner of the experimental cup differed slightly from those of the simulated cup, but the difference again remained within acceptable engineering tolerance. The experimental cup's wall was identical to the simulated cup's wall, with the difference being less than 0.05 mm at the top of the cup. For $r_{90°}$, the bottom of the experimental cup was similar to that of the simulated cup; the difference was again within acceptable engineering tolerance. The experimental cup's wall, as well as the top portion, was almost identical to the simulated cup's wall. The optimal match was achieved at $r_{90°}$.

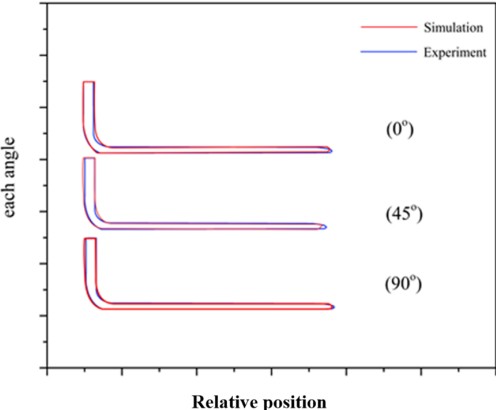

**Figure 11.** Comparison of experimental and simulated cup shapes (Case D).

3.1.3. Comparison of Cup Thickness

The cups cut along angles $r_{0°}$ (Figure 13a), $r_{45°}$ (Figure 13b), and $r_{90°}$ (Figure 13c) were compared with a simulated cup created in DEFORM. The cups were cut using a wire-cutting electrical discharge machine; rinsed using an ultrasonic cleaner; embedded using an automatic mounting press; ground and polished using a metallographic grinding and polishing machine; and evaluated using white light interferometry, a laser displacement meter, and an image-measuring instrument. For $r_{0°}$, a 0.1 mm difference was observed

at the bottom of the cup, but this difference was within acceptable engineering tolerance. The rounded corner and cup's wall were almost identical to their simulated counterparts. For $r_{45°}$, a 0.15 mm difference was discovered at the cup's bottom, whereas a 0.05 mm difference was observed at the rounded corner, but these were again within acceptable engineering tolerance. The difference along the cup wall was negligible. For $r_{90°}$, a 0.1 mm difference, which was within acceptable engineering tolerance, was discovered at the cup's bottom. The differences at the rounded corner and along the cup wall were negligible. The optimal match was achieved at $r_{90°}$.

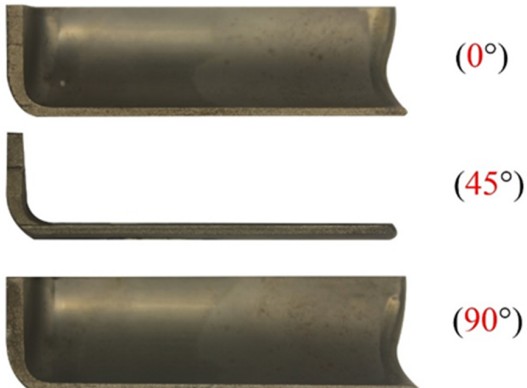

**Figure 12.** Cups cut along angles $r_{0°}$, $r_{45°}$, and $r_{90°}$.

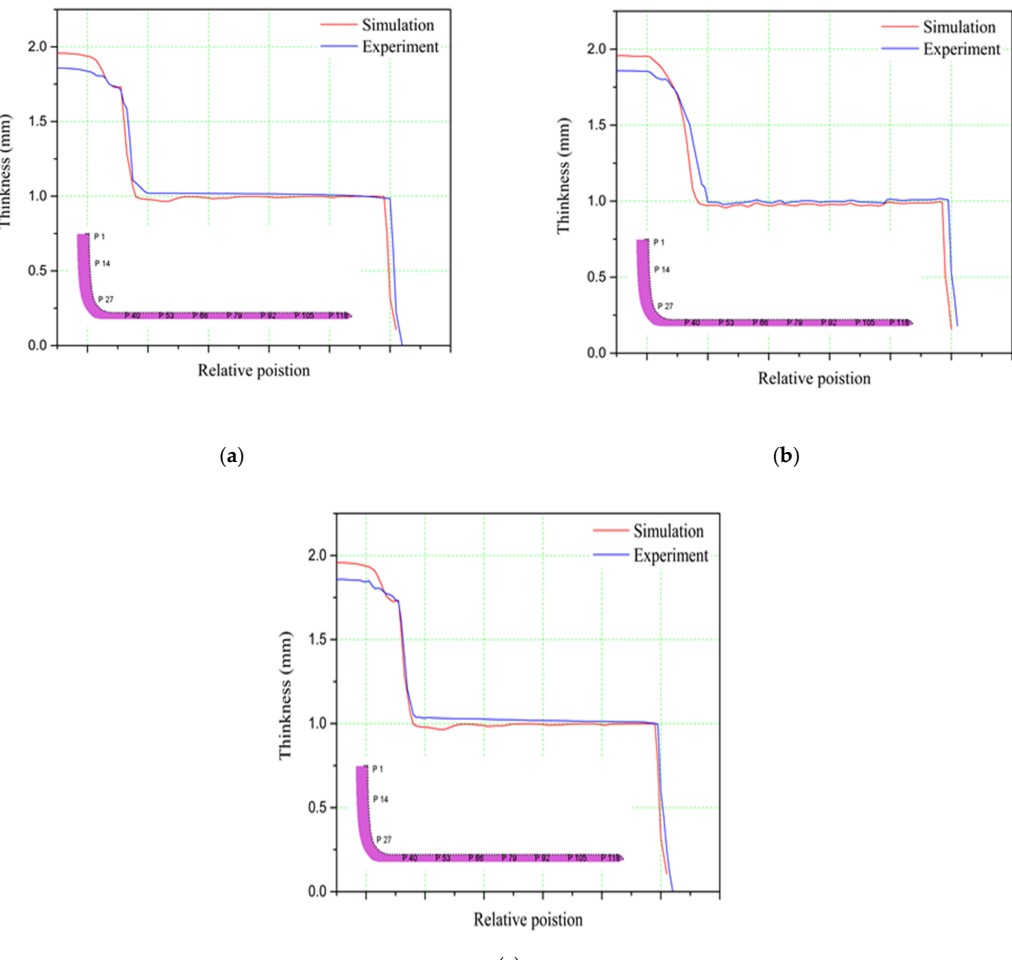

**Figure 13.** Thickness distribution along angle (**a**) 0°; (**b**) 45°; (**c**) 90°.

### 3.2. Extrusion Parameter Analysis: Establishment of Punch Motion Curves

This section analyzes changes in punch motion curves. Three punch motion curves—Cases A–C—were established, and these all followed the conventional single-stroke punch motion pattern. In Case A, the punch moved down until the end of the extrusion process (Figure 14a). Case B and Case C differed from Case D in that the servo vibration frequency was increased. Specifically, in Case B, the punch moved down for 1 mm and then moved up 0.5 mm (Figure 14b), whereas in Case C, the punch moved down for 2 mm and then moved up 0.5 mm (Figure 14c). The differences between these curves and the conventional punch motion curve are presented in Table 7. The punch motion curves' relationships with the cup's height and thickness, the extrusion force–displacement curve, stress, and strain were analyzed. The blanks used for circular cup extrusion were all 2 mm thick, with a coefficient of friction m of 0.1, and made of an anisotropic material.

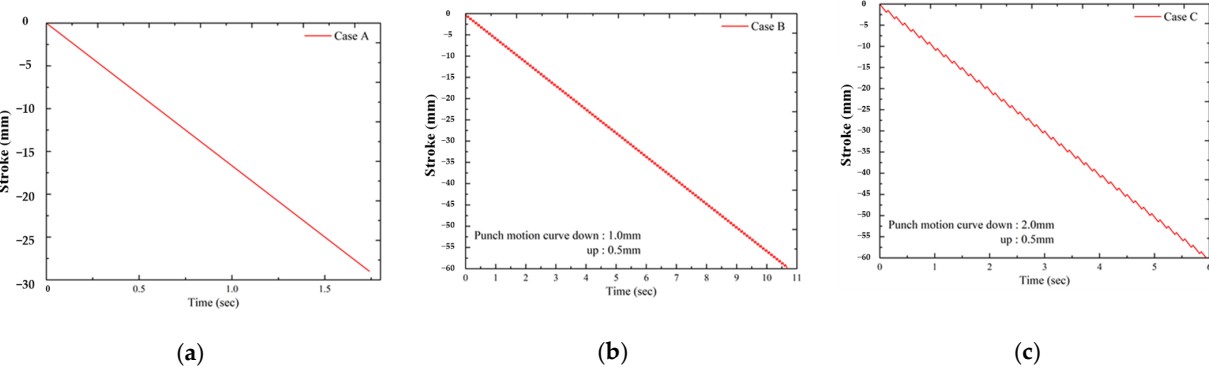

(**a**)  (**b**)  (**c**)

**Figure 14.** (**a**) Conventional punch motion curve (Case A); (**b**) Punch motion curve for 1 mm punch movement (Case B); (**c**) Punch motion curve for 2 mm punch movement (Case C).

**Table 7.** Differences between punch motion curves.

| Motion Curve / Type | Pattern | Stroke Difference |
|---|---|---|
| Case A | | All the way down |
| Case B | | Down 1.0 mm Up 0.5 mm |
| Case C | | Down 2.0 mm Up 0.5 mm |
| Case D | | Down 3.0 mm Up 0.5 mm |

### 3.3. Effect of Punch Motion Curve on Circular Cup Extrusion

This section discusses the effect of the punch motion curve on the extrusion process. The relationships of the punch motion curves with the cup's height and thickness, extrusion force–displacement curve, stress, and strain were analyzed with the stroke set at 60 mm.

3.3.1. Extrusion Force–Displacement Curves

Figure 15 displays the extrusion force–displacement curves for the conventional and experimental punch motion patterns. The figures indicate that a greater extrusion force was required in Case A than in the other cases. It also reveals that when the punch moved upward, the stress on the blank instantly disappeared. Moreover, the smallest extrusion force

was required in Case B because the punch was released more frequently. By contrast, in Case D, the punch moved downward for 3 mm, resulting in larger stress and strain. In the second round, a greater extrusion force was required. This caused the blank to become harder and more brittle, which in turn resulted in a shorter cup. In Case B, the strain hardening was less pronounced because of the smaller stroke and greater frequency. The blank was thus more stretchable, resulting in a taller cup. The differences in the maximum extrusion force between the conventional and experimental punch motion curves are detailed in Table 8.

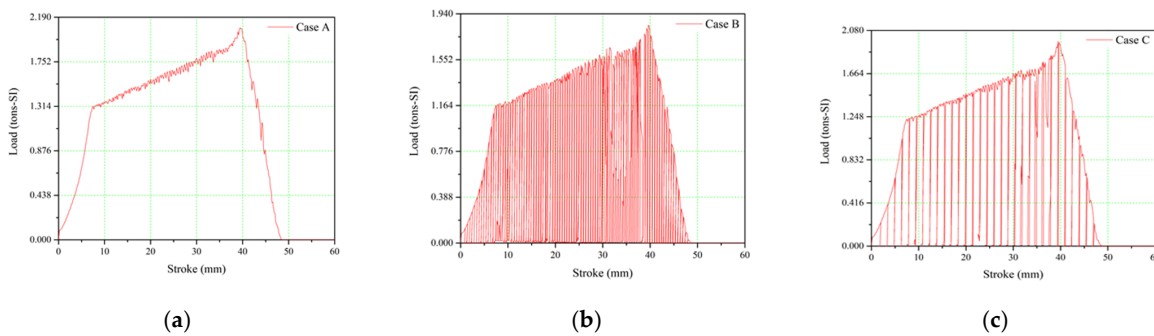

| (a) | (b) | (c) |

**Figure 15.** Extrusion force–displacement curve of (**a**) Case A; (**b**) Case B; (**c**) Case C.

**Table 8.** Differences in maximum extrusion force between conventional and experimental punch motion curves.

| Cup / Motion Curve | Load (tons-SI) | Difference % |
|---|---|---|
| Case A | 2.09 | 0.00 |
| Case B | 1.83 | −12.44 |
| Case C | 1.96 | −6.22 |
| Case D | 2.02 | −3.35 |

### 3.3.2. Comparison of Cup Height

The cup height differences between the conventional punch motion curve (Case A) and experimental punch motion curves (Cases B, C, & D) were investigated. As illustrated in Figure 16, the cup height was 43.5008 mm for Case A, 44.4263 mm for Case B, 44.2160 mm for Case C, and 44.0235 mm for Case D. The punch motion curve in Case B thus resulted in a considerably greater cup height than the other curves; percentage calculation of differences were 1.20%~2.12% compared with the height for Case A (Table 9).

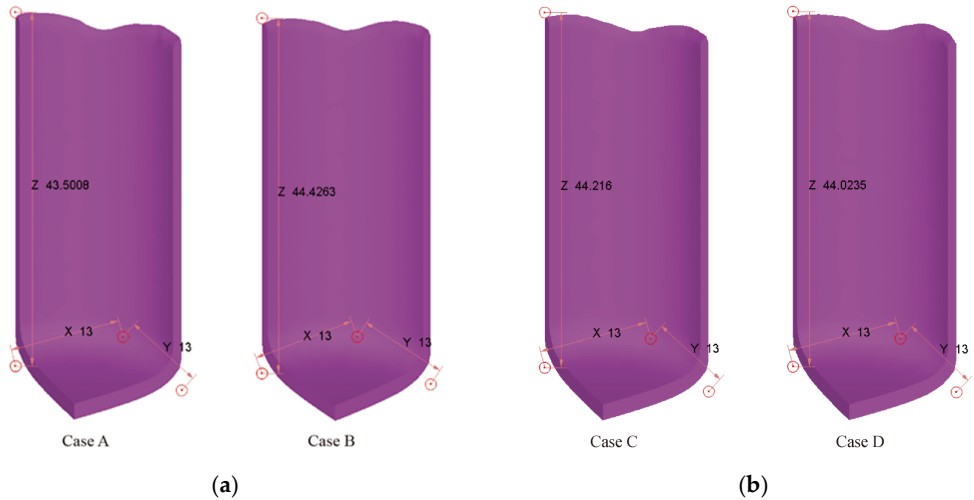

| (a) | (b) |

**Figure 16.** Cup height in (**a**) Cases A and B; (**b**) Cases C and D.

**Table 9.** Cup height resulting from conventional and experimental punch motion curves.

| Motion Curve / Cup | Height (mm) | Difference % |
|---|---|---|
| Case A | 43.5008 | 0.00 |
| Case B | 44.4263 | 2.12 |
| Case C | 44.2160 | 1.64 |
| Case D | 44.0235 | 1.20 |

### 3.3.3. Comparison of Cup Thickness

A circular cup was cut in half using a wire-cutting electrical discharge machine, and its thickness was then measured from the bottom to the top. The thickness distribution of the cup cut along the 0°, 45°, and 90° angles is displayed in Figure 17. The punch motion curve in Case A resulted in the greatest thickness at the rounded corner. By contrast, Case B resulted in the smallest thickness at the rounded corner. Cases B and A resulted in the most and least even cup wall thickness, respectively. Overall, the optimal extrusion performance was achieved using servo press.

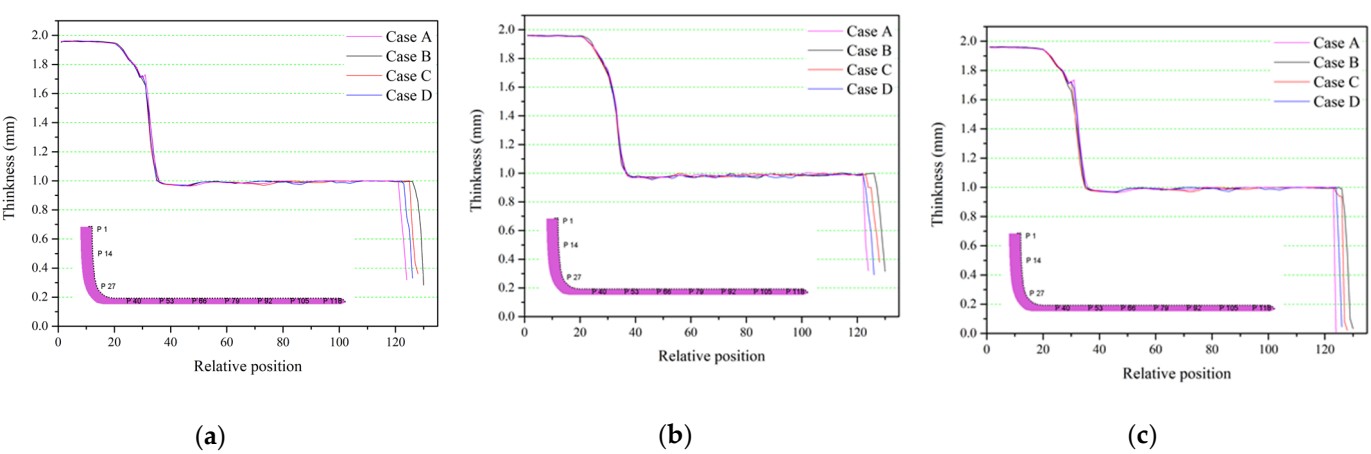

(a)　　　　　　　　　　(b)　　　　　　　　　　(c)

**Figure 17.** Thickness distribution along angle (**a**) 0°; (**b**) 45°; (**c**) 90°.

### 3.3.4. Stress and Strain Distribution

The stress and strain diagrams of the conventional punch motion curve (Case A) and experimental punch motion curves (Cases B–D) are presented in Figure 18. Case A produced the greatest stress. Because stress was effectively released when the punch moved upward in Cases B–D, the simulation results for these cases indicated that the cup was subjected to less stress during the extrusion process. However, the strain upon the cup did not differ much from that experienced in Case A. Table 10 presents the difference in maximum stress values.

**Table 10.** Difference in maximum stress.

| Motion Curve / Cup | Stress(MPa) | Difference (%) |
|---|---|---|
| Case A | 2560 | 0.00 |
| Case B | 862 | −66.33 |
| Case C | 1010 | −60.55 |
| Case D | 1310 | −48.83 |

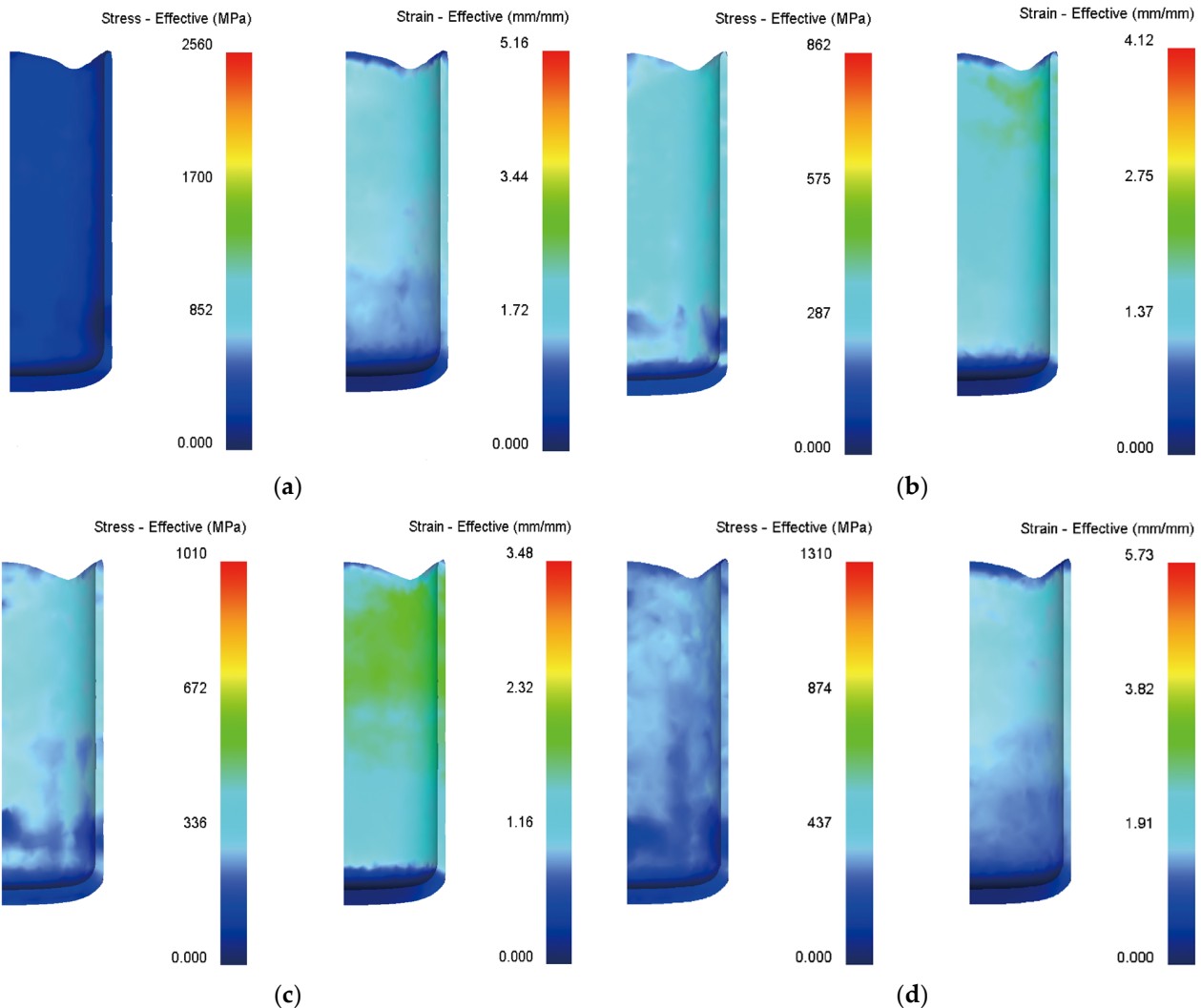

**Figure 18.** Stress and strain distribution induced by (**a**) Case A (**b**) Case B (**c**) Case C (**d**) Case D.

*3.4. Effect of Coefficient of Friction on the Extrusion Process*

The effect of the coefficient of friction in Cases A–D was investigated. Three coefficient of friction values (m = 0.05, 0.3, & 0.5) were employed in the extrusion of an anisotropic material; the thickness of the blank was set to 2 mm and the stroke to 60 mm.

3.4.1. Relationship between Coefficient of Friction and Cup Height

Table 11 presents the cup heights at various coefficients of friction. Figure 19 illustrates the relationship of the coefficient of friction with the cup height and cup height difference, respectively. The coefficient of friction had the weakest effect on extruded cup height when it was 0.05 and the strongest effect when it was 0.5, indicating that the coefficient of friction did indeed have an effect.

**Table 11.** Cup heights at various coefficients of friction.

| Cup Motion Curve | m = 0.05 | | m = 0.3 | | m = 0.5 | |
|---|---|---|---|---|---|---|
| | Height (mm) | Difference | Height (mm) | Difference | Height (mm) | Difference |
| Case A | 43.4452 | 0.00 | 44.0505 | 0.00 | 44.6912 | 0.00 |
| Case B | 44.2696 | 1.90 | 45.3005 | 2.37 | 45.8017 | 2.48 |
| Case C | 44.0688 | 1.43 | 44.9451 | 2.03 | 45.6628 | 2.17 |
| Case D | 43.9069 | 1.06 | 44.7284 | 1.53 | 45.4447 | 1.69 |

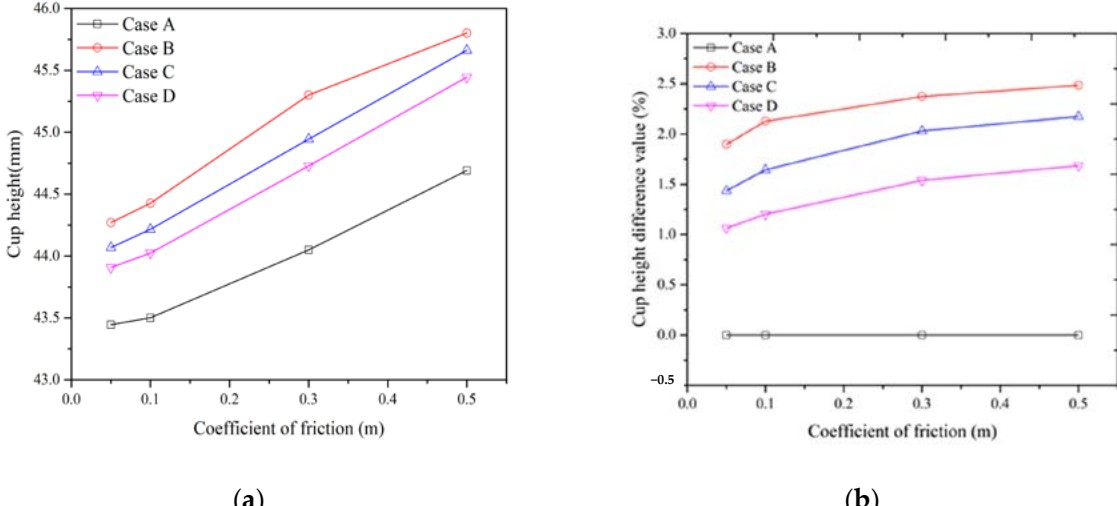

(**a**)                                                  (**b**)

**Figure 19.** Relationship between coefficient of friction and (**a**) cup height; (**b**) cup height difference.

### 3.4.2. Relationship between Coefficient of Friction and Stress

Table 12 details the stress at various coefficients of friction. Figure 20 shows the relationships of the coefficient of friction with stress and the stress difference, respectively. The stress was low when m = 0.05 and high when m = 0.5. This suggests that stress increased with the coefficient of friction.

**Table 12.** Stress at various coefficients of friction.

| Cup Motion Curve | m = 0.05 | | m = 0.3 | | m = 0.5 | |
|---|---|---|---|---|---|---|
| | Stress (MPa) | Difference (%) | Stress (MPa) | Difference (%) | Stress (MPa) | Difference (%) |
| Case A | 1650 | 0.00 | 4010 | 0.00 | 6260 | 0.00 |
| Case B | 822 | −50.18 | 1440 | −64.09 | 2460 | −60.70 |
| Case C | 980 | −40.60 | 2340 | −41.65 | 3760 | −39.94 |
| Case D | 1060 | −35.76 | 2850 | −28.93 | 4820 | −23.00 |

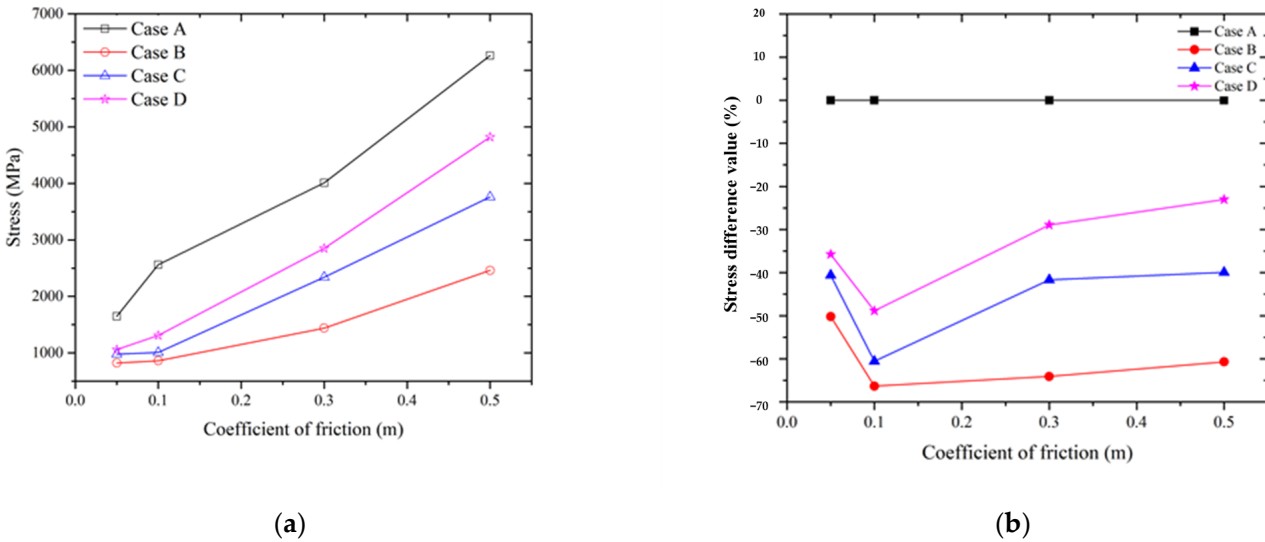

(**a**)                                                  (**b**)

**Figure 20.** Relationship between coefficient of friction and (**a**) stress; (**b**) stress difference.

## 4. Conclusions

Industrial experience indicates that the punch motion curve of a servo press usually results in higher extrusion performance than the curve of a conventional stamping press. Moreover, servo presses can create cups of greater height. This study verified that a punch motion curve can be applied in metal shaping to reduce the spring back of the sheet metal, reduce the stress in the metal, and improve its formability. The following conclusions are drawn from the discussion presented in previous sections:

(1). The method developed in this study can be applied in the extrusion–drawing process when creating circular cups and when analyzing the spring back of U-shaped sheet metal. The differences noted in this study were all within acceptable engineering tolerance.

(2). Because SPCG sheets are designed for deep drawing, they are used to extrude circular cups without any lubricant.

(3). For the influence of extrusion force–displacement, a greater extrusion force was required in Case A (conventional punch motion) than in the other cases. Moreover, the smallest extrusion force was required in Case B because the punch was released more frequently, and the strain hardening was also less pronounced. The blank was thus more stretchable, resulting in a taller cup.

(4). For the comparison of cup height, the cup height was 44.4263 mm for Case B and resulted in a greatest cup height than the other cases. The height difference between case B and case A (conventional punch motion) is 2.12%.

(5). For the comparison of cup thickness, Case A resulted in the greatest thickness at the rounded corner. By contrast, Case B resulted in the smallest thickness at the rounded corner. So, the optimal extrusion performance was achieved using servo press.

(6). In this study, the optimal balance between cup shape and thickness distribution was achieved using the combination of Case D and $r_{90°}$.

**Author Contributions:** Conceptualization, T.-C.C.; methodology, T.-C.C.; software, S.-X.C.; validation, T.-C.C., S.-X.C. and C.-C.W.; formal analysis, T.-C.C. and S.-X.C.; investigation, T.-C.C. and S.-X.C.; writing—original draft preparation, T.-C.C., S.-X.C. and C.-C.W.; writing—review and editing, C.-C.W.; visualization, C.-C.W.; supervision, C.-C.W.; project administration, C.-C.W.; funding acquisition, T.-C.C. All authors have read and agreed to the published version of the manuscript.

**Funding:** This research was funded by Ministry of Science and Technology in Taiwan, grant number MOST 109-2221-E-167-007.

**Data Availability Statement:** Not applicable.

**Conflicts of Interest:** The authors declare no conflict of interest.

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
