# Peer review of "Punch Motion Curve in the Extrusion–Drawing Process to Obtain Circular Cups"

_machines, doi:10.3390/machines10080638_

Round 1

Reviewer 1 Report

The authors have conducted research and made some interesting points related to the Punch Motion Curve in the Extrusion–Drawing Process. I think the following issues can be explored.

\1. It is recommended that the authors include a table in the introduction to demonstrate the strengths and weaknesses of traditional methods visually.

\2. It is suggested that the authors add a description of the comparison between the results of this paper and traditional research, as well as a summary of this work at the end of the introduction.

\3. It is suggested that the authors add a few more references to recent articles by other researchers in the field.

\4. It is recommended that the authors compare the appropriate parameters with the most current research to highlight the strengths of the approach presented in this paper. 

Author Response

Responses :

The authors appreciate reviewers’ valuable comments very much. Modifications have been made to improve the manuscript according to reviewers’ suggestions as summarized below:

  • It is recommended that the authors include a table in the introduction to demonstrate the strengths and weaknesses of traditional methods visually. Please refer the revised paper in page 2.

Response: According to the reviewer’s comments, we include a table (Table 1) in the introduction to demonstrate the strengths and weaknesses of traditional methods visually.

Table 1. The strengths and weaknesses of traditional methods

strengths

weaknesses

traditional methods

Fast processing speed

1.         single-stroke stamping presses

2.         only one punch motion curve and the motion profiles can’t be adjusted

3.         only move at a fixed speed for sheet metal stamping

4.         wall thickness of the finished product is extremely uniform

  • It is suggested that the authors add a description of the comparison between the results of this paper and traditional research, as well as a summary of this work at the end of the introduction.

Response:

Thank you very much for your comments. We have added a description of the comparison between the results of this paper and traditional research, as well as a summary of this work at the end of the introduction. Please refer the revised paper in pages 3-4. This paragraph is shown as following:

In the present study, the servo curve method is proposed to show that as the frequency of the punch motion curve is the highest, a higher cup height can be obtained. The difference in the height of the cup during the extrusion-drawing process is only a slight change of 1.20%~2.12%. For the distribution of the thickness of the circular cup, the cup wall produced by the servo curve is the most uniform. If the cup wall obtained by the traditional curve is quite uniform, the servo curve method proposed in this paper will obtain the best extrusion-drawing effect. In this study, the circular cup extrusion-drawing process is designed according to the punch motion curve. In the future, the circular cup extension-drawing simulation analysis can be carried out, and a complete database of the circular cup extension-drawing process can be established to provide industrial production use to improve its production efficiency.

  • It is suggested that the authors add a few more references to recent articles by other researchers in the field.

Response:

Thank you very much for your comments and we have added three relative references in recently 2 years. Please refer the revised paper in pages 3-4 and 21. This paragraph is shown as following:

In 2020, Kuo et. al., studied the SUS304 rectangular cup stamping and optimized the pulsating curve for a servo press by finite element method. Meanwhile, they applied the Taguchi method to obtain the optimal parameter combinations and the optimization results showed that a shorter forming time (0.06 s less), a lower thinning ratio lower (0.1425 % less), and a smaller forming force (808 N smaller) [12]. Kriechenbauer et. al. proposed a systematic design of deep-drawing processes with free force and motion functions on servo presses based on computational science methods. They determined the optimal parameters (force and motion functions) for a deep-drawing process with superimposed vibrations on servo screw presses. The results of the evolutionary optimization approaches are validated by experiments with cross die part [13]. Choudhari et. al. used the numerical and experimental approaches to analyze the effect of different drawing parameters such as blank shape, blank thickness, load, dry/wet lubrication on square cup drawing process for extra deep drawn steel sheet material. Simulation results are validated through experimentation. The optimized process parameters can be formed a square cup without any defects such as thinning, wrinkling, etc. The results showed that for considered process parameters, formability of material having a blank thickness of 2 mm is better as compared to a blank thickness of 1 mm and 0.8 mm, for load of 100 kN with dry lubrication [14].

  • It is recommended that the authors compare the appropriate parameters with the most current research to highlight the strengths of the approach presented in this paper.

Response:

Thank you very much for your suggestions. It is very important to compare the appropriate parameters with the most current research to highlight the strengths of the approach presented in this paper. So, in this paper, we analyzed the circular cup extrusion–drawing process to propose three punch motion curve of a servo press (Cases B–D) and compare with a conventional stamping press (Case A). The strokes of Cases B to D differed from that of Case A by 0.5, 1.5, and 2.5 mm, respectively were analyzed and compared, particularly regarding the effect of the coefficient of friction on the circular cup extrusion-drawing process. Accordingly, we have highlighted the strengths of proposed method in Section 4 (Conclusions) of the revised paper.

Please refer the revised paper in page 19.

Industrial experience indicates that the punch motion curve of a servo press usually results in higher extrusion performance than the curve of a conventional stamping press. Moreover, servo presses can create cups of greater height. This study verified that a punch motion curve can be applied in metal shaping to reduce the spring back of the sheet metal, reduce the stress in the metal, and improve its formability. The following conclusions are drawn from the discussion presented in previous sections:

  • The method developed in this study can be applied in the extrusion–drawing process when creating circular cups and when analyzing the spring back of U-shaped sheet metal. The differences noted in this study were all within acceptable engineering tolerance.
  • Because SPCG sheets are designed for deep drawing, they are used to extrude circular cups without any lubricant.
  • For the influence of extrusion force–displacement, a greater extrusion force was required in Case A (conventional punch motion) than in the other cases. Moreover, the smallest extrusion force was required in Case B because the punch was released more frequently and the strain hardening was also less pronounced. The blank was thus more stretchable, resulting in a taller cup.
  • For the comparison of cup height, the cup height was 44.4263 mm for Case B and resulted in a greatest cup height than the other cases. The height difference between case B and case A (conventional punch motion) is 2.12%.
  • For the comparison of cup thickness, Case A resulted in the greatest thickness at the rounded corner. By contrast, Case B resulted in the smallest thickness at the rounded corner. So, the optimal extrusion performance was achieved using servo press.
  • In this study, the optimal balance between cup shape and thickness distribution was achieved using the combination of Case D and .

Reviewer 2 Report

The article deals with interesting technological issues. It is written very well in terms of content. As the authors write, servo press technology is gaining in popularity because its punch motion curve offers greater deformability than a conventional stamping press. The authors analyzed the shape of the punch and the curves of movement. They describe different shapes. The DEFORM simulation was used. The introduction is correctly written. The literature review is good. It does not require correction. The research part, the description of the method, solutions and model is very good. All the introduced illustrations do not contain errors. They are well introduced and well-founded. The discussion part is well-prepared, thought-out and perfectly describes the intentions of the authors. The summary includes the most important conclusions in points. Numerical values ​​have also been introduced. The work is quite extensive, but all the materials introduced are consistent. I do not see any major disadvantages of the article. In my opinion, the article can be accepted.

Author Response

The authors appreciate reviewers’ valuable comments very much.

Reviewer 3 Report

There are some corrections necessary:

Rows 131-132 move before 139;

Fig. 4b, 5a, 5b, 6a, 6b are not necessary.

Fig. 13 - horizontal axe !

Row 273 !

Fig. 20 !

Row 333 !

Row 338 !

Author Response

Response:

  1. The Caption of Figure 3 has been revised in the correct position and please refer the revised paper in page 6.
  2. According to the reviewer’s comment, we have removed the Figs. 4b, 5a, 5b, 6a, 6b and please refer the revised paper in pages 6-8.
  3. The description of the horizontal axis in Fig.13 is revised and please refer the revised paper in page 11.
  4. In Row 273, we have revised the “statistical analysis….” to “percentage calculation of differences….” and please refer the revised paper in page 14.
  5. In Fig. 20, we have modified all the figures and please refer the revised paper in pages 16-17.
  6. In Row 333, we have revised the “…no lubricant is required…” as “…they are used to extrude circular cups without…” and please refer the revised paper in page 19.
  7. In Row 338, we have modified “r_(90°)” as “ ” and please refer the revised paper in page 19.

Round 2

Reviewer 1 Report

The revision is satisfactory.